# Aggrecan, the Primary Weight-Bearing Cartilage Proteoglycan, Has Context-Dependent, Cell-Directive Properties in Embryonic Development and Neurogenesis: Aggrecan Glycan Side Chain Modifications Convey Interactive Biodiversity

**DOI:** 10.3390/biom10091244

**Published:** 2020-08-27

**Authors:** Anthony J Hayes, James Melrose

**Affiliations:** 1Bioimaging Research Hub, Cardiff School of Biosciences, Cardiff University, Cardiff CF10 3AX, UK; HayesAJ@cardiff.ac.uk; 2Raymond Purves Laboratory, Institute of Bone and Joint Research, Kolling Institute of Medical Research, Northern Sydney Local Health District, Royal North Shore Hospital, St. Leonards, NSW 2065, Australia; 3Graduate School of Biomedical Engineering, University of New South Wales, Sydney, NSW 2052, Australia; 4Sydney Medical School, Northern, The University of Sydney, Faculty of Medicine and Health at Royal North Shore Hospital, St. Leonards, NSW 2065, Australia

**Keywords:** aggrecan, tissue morphogenesis, HNK-1 trisaccharide, glycosaminoglycan, cellular regulation, extracellular matrix

## Abstract

This review examines aggrecan’s roles in developmental embryonic tissues, in tissues undergoing morphogenetic transition and in mature weight-bearing tissues. Aggrecan is a remarkably versatile and capable proteoglycan (PG) with diverse tissue context-dependent functional attributes beyond its established role as a weight-bearing PG. The aggrecan core protein provides a template which can be variably decorated with a number of glycosaminoglycan (GAG) side chains including keratan sulphate (KS), human natural killer trisaccharide (HNK-1) and chondroitin sulphate (CS). These convey unique tissue-specific functional properties in water imbibition, space-filling, matrix stabilisation or embryonic cellular regulation. Aggrecan also interacts with morphogens and growth factors directing tissue morphogenesis, remodelling and metaplasia. HNK-1 aggrecan glycoforms direct neural crest cell migration in embryonic development and is neuroprotective in perineuronal nets in the brain. The ability of the aggrecan core protein to assemble CS and KS chains at high density equips cartilage aggrecan with its well-known water-imbibing and weight-bearing properties. The importance of specific arrangements of GAG chains on aggrecan in all its forms is also a primary morphogenetic functional determinant providing aggrecan with unique tissue context dependent regulatory properties. The versatility displayed by aggrecan in biodiverse contexts is a function of its GAG side chains.

## 1. Introduction

A vast amount has been written over the last five decades on aggrecan’s structure (Figure 1) and function in weight-bearing connective tissues such as hyaline cartilage and intervertebral disc (IVD) in health and disease [1,2,3,4,5,6,7,8]. However, aggrecan also has important roles in tensional connective tissues (e.g., meniscus, tendon and ligaments) [9,10] as well as in non-cartilaginous tissues such as the heart and nervous system [11,12,13,14,15,16,17,18,19,20,21,22]. While aggrecan has important roles in tissue development and function, surprisingly little has been published on its interactive and cell-directive properties in tissue morphogenesis. This review aims to rectify this deficiency but cannot be meaningfully undertaken without first covering aggrecan’s functional attributes in weight-bearing tissues that contribute to matrix stabilisation. This diversity in aggrecan’s functional properties is due to modifications in its glycosaminoglycan (GAG) side chains which equip it with unique ligand interactivity in specific developmental contexts.

## 2. CS Sulphation on Aggrecan Is an Important Functional Determinant

Studies have shown that two out of every seven non-reducing termini of normal [23] and chondrosarcoma [24] aggrecan CS chains contain 4, 6-disulphated GalNAc. Non-reducing terminal GalNAc4S or GalNAc4,6S can be linked to either a 4-sulphated or a 6-sulphated disaccharide. In a further study, CS from juvenile and adolescent growth plate cartilage was shown to contain non-reducing terminal GalNAc4S, whereas in adult cartilages approximately half of the non-reducing termini were disulphated GalNAc4,6S [25], representing an increase in aggrecan sulphation with tissue maturation. It is these sulphate groups which provide aggrecan’s interactive properties with a number of ligands; a high density of sulphate and carboxylate groups in aggrecan confer its remarkable ability to imbibe water and to provide tissue hydration that allows some tissues to withstand compressive loading (Figure 1a,f).

While clear functional roles for HS-PGs in cell signalling transduction pathways are well established, roles for CS-PGs in such processes have often been given lower importance; nevertheless, CS can also modulate cell-signalling pathways involving hedgehog proteins, wingless-related proteins and fibroblast growth factors [26,27,28,29,30,31,32,33,34,35]. Indeed, the co-distribution of these components with aggrecan in growth plate cartilages and localisation of particular CS sulphation motifs with chondroprogenitor cell populations associated with diarthrodial joint development (Figure 2a,f) alludes to multiple interactive possibilities [36]. Studies with brachymorphic mice, nanomelic chick, dyschondroplastic chicken and Cmd mutant mice clearly show the importance of aggrecan in growth plate cartilage development and skeletogenesis. Furthermore, individuals suffering from Kashin–Beck disease, an endemic osteochondropathy that occurs in certain parts of China, is characterised by small stature and deformities of the limbs and digits, distorted growth plates, chondrocyte apoptosis and low levels of aggrecan [37,38,39,40,41]. The correct sulphation of the CS chains of aggrecan is essential to generate functional determinants capable of interacting with growth factors and morphogens [42,43,44,45]. Approximately one in three of aggrecan’s CS chains have a non-reducing terminal chondroitin-4, 6 disulphated residue (CS-E) in articular cartilage [46]. Highly sulphated CS-E binds the HS binding growth factors midkine and pleiotrophin [47,48,49].

Approximately two in seven CS chains are terminated in 4, 6 disulphated GalNAc, which varies with the age and cartilage type; four in seven of CS chains are terminated by 4-sulphated GalNAc; and one in seven CS chains are terminated in a GlcUA linked to 4-sulphated GalNAc. Non-reducing terminal 4,6-disulphated GalNAc residues are 60-fold more abundant than 4,6-disulphated GalNAc in interior regions of the CS chain [24].

CS chains terminated in 4-sulphated GalNAc predominate in aggrecan from foetal to 15-year-old knee cartilage, whereas, in 22–72-year-olds, 50% of the CS chains were terminated in 4,6-disulphated GalNAc. GlcUA-4-sulphated GalNAc disaccharides terminated 7% of CS chains in foetal to 15-year-old cartilage but fell to 3% in adults, whereas GlcUA-6-sulphated GalNAc represented 9% of the CS chains in foetal to 72-year-old cartilage. This disaccharide is recognised by MAb 3-B-3 (−) [46].

The distribution of 4- and 6-sulphated CS epitopes is variable along a CS chain in aggrecan and is influenced by the maturational status of the cartilage or the extent to which the cartilage was sampled from a high or low weight-bearing cartilage region [3]. Certain trends have been observed in the sulphation patterns of CS in aggrecan chains. C-4-S is more predominant in aggrecan from foetal and young articular cartilage and occupies a central region in the CS chain, whereas non-sulphated chondroitin is more predominant towards the linkage region. C-6-S has a predominant distribution towards the non-reducing terminus and is more abundant in mature cartilage to the detriment of C-4-S sulphation [51].

Graded partial digestions of CS chains with chondroitinase ABC or ACII reveals regions along the CS chain where MAbs 6C3, 4C3 and 7D4 are most immunoreactive [51]. MAb 6C3 reacts optimally with regions of CS chains towards the non-reducing terminus where C-6-S predominates, and this reactivity is removed during early stages of chondroitinase digestion. Further digestion removes MAb 4C3 reactivity and continued digestion then removes reactivity to MAb 7-D-4. While the specific epitopes identified by MAb 4C3 and 7D4 are yet to be identified, reactivity of these antibodies in a range of tissues undergoing morphogenetic transition during development displays subtly different immunolocalisation patterns and are of functional significance [52,58,59,60,61,62,63,64,65]. 

MAb 3-B-3 identifies a non-reducing terminal disaccharide in CS consisting of GlcUA-GalNAc-6-sulphate, which is termed a 3-B-3(−) epitope to distinguish it from the 3-B-3(+) stub epitope disaccharide attached to the linkage region that is generated by exhaustive end-point digestion of CS chains by chondroitinase ABC [64]. As noted above, this non-reducing terminal 3-B-3(−) epitope occurs in approximately two in every seven CS chains; disulphated C-4,6-S and C-6-S GalNAc also occur as components in this non-reducing terminal disaccharide in CS chains [24,46].

## 3. HNK-1 Aggrecan Regulates Neural Crest Cell Migration during Embryonic Development

Neural crest stem cells (NCSCs) are a transient multipotent migratory embryonic neuroepithelial cell population present in vertebrate embryos [66]. The neural crest (NC) gives rise to the neural tube and notochord, neurons and glia of the peripheral nervous system/central nervous system (PNS/CNS), melanocytes, cartilaginous and bony tissues of the craniofacial skeleton as well as cephalic neuroendocrine organs and some cardiac tissues, including large vessels, valve leaflets and heart tendons. NCSCs express Sox 10 and HNK-1 and transition to a mesenchymal NC tissue during early embryonic development [67]. HNK-1 is a highly interactive functional module participating in homophilic and heterophilic interactions [68] in a number of neural PGs and cell adhesive proteins [69]. HNK-1 is also widely expressed on a number of myelin-associated glycoproteins such as L1, myelin-associated glycoprotein (MAG), TAG-1 (transient axonal glycoprotein) [70] and P0 as well as sulpho-glucuronyl glycolipids (SGGLs, SGGL-1 and SGGL-2) which have important roles to play in the remyelination of damaged axons [69]. P0 and MAG are integral transmembrane glycoprotein components of peripheral nerve myelin (Figure 3a,d). TAG-1, a GPI anchored 135-kDa glycoprotein expressed transiently on the surface of subsets of neurons in the developing mammalian nervous system, has neurite outgrowth promoting activity [70]. HNK-1 also mediates neural cell attachment to laminin in ECM structures [71,72]. During early embryonic development, HNK-1 decorates aggrecan in the notochord, and this form of aggrecan has roles in the directional control of NCSCs in the development of the neural tube, notochord, neural networks and associated tissues (Figure 3e).

## 4. Variation in the CS Chain Fine Structure with Development and Pathology in Health and Disease

Several years ago [58,64,73], MAbs 3-B-3(−) and 7-D-4 were shown to identify chondrocyte-clusters in pathological (osteoarthritic) canine and human articular cartilage. At that time, which pre-dated knowledge of stem/progenitor cell niches in tissues, these cell-clusters were considered a classical feature of the onset of late stage degenerative joint disease and were interpreted to indicate a failed, late-stage, response to replace PGs in a matrix extensively degraded by matrix proteases. An alternative explanation of this cellular phenomenon however has now emerged. It is now believed that these “chondrocyte clusters” arise from adult stem/progenitor cell niches [74,75,76,77]. The 3-B-3(−), 4-C-3 and 7-D-4 CS sulphation motifs also occur in foetal development and are markers of anabolic processes in transitional tissues (Figure 2a,d). An important feature of the stem/progenitor cell niche is the sulphation of the PG GAG side chains (Figure 1a,f). Variable expression of GAG sulphotransferases and glycosyl transferases in stem/progenitor cell niches (Figure 2a,c,d) supports such an interpretation [78,79,80]. Cell clusters have also been shown to express Notch 1 and CD166, biomarkers that are synonymous with the stem cell niche [74,81].

## 5. Effects of Modulation of CS Sulphation on Gene Expression and Cartilage Development

In chondrocyte cultures, p-nitrophenyl xyloside (PNPX) acts as a competitive acceptor of CS/DS substitution on PG core proteins [82]. PNPX treatment reduces SOX-9, aggrecan and collagen type II gene expression, levels of collagen type II protein synthesis and PG sulphation. It also leads to delayed expression of native CS/DS sulphation motif epitopes and delayed chondrogenic differentiation of bovine MSCs accompanied with reduced tissue development. While the precise role of native CS sulphation motifs identified by MAb 3-B-3(−), 4-C-3, 7-D-4 and 6-C-3 in transitional tissues are not known, they appear to be of importance in the initial stages of chondrogenesis and their distribution patterns indicate they have roles in morphogenetic signalling through the capture and cellular presentation of soluble bioactive molecules (growth factors, morphogens, etc.) of importance in tissue development and morphogenesis [51,58,59,60,83] (Figure 2a,c,d).

## 6. Aggrecans Roles in Articular Cartilage, Fibrocartilages, Heart and Neural Tissues

Aggrecan is a large KS and CS substituted lectican PG family member with important space filling and water imbibing properties. In weight-bearing articular cartilages aggrecan forms macro-aggregate structures through interaction of its N-terminal G1 domain with hyaluronan and link protein [2,5,6]. Aggrecan–HA aggregates have important water-imbibing properties that entrap water in tissues in a dynamic manner. These properties allow the aggrecan-rich tissues to resist compression and equips articulating tissues in synovial joints with their weight-bearing properties. Cartilage is also self-lubricating through moisture expelled at the cartilage surface when the joint is loaded arising from aggrecan associated water molecules. This is a dynamic process with moisture returning to the cartilage when load is reduced or removed from the joint. Aggrecan is widely distributed in the articular hyaline cartilages of diarthrodial joints, but also occurs in the elastic and fibrocartilages of rib, nasal and tracheal cartilages, larynx, outer ear and the epiglottis [84,85,86,87]. Aggrecan is also important in foetal heart development and is a functional ECM component, which contributes to the resilience of the endocardium, myocardium, epicardium and valve leaflets of mature heart tissue [17,88]. Aggrecan is also found in the CNS and PNS in perineuronal net (PNNs) structures. These are aggrecan–HA–tenascin C aggregate structures which localise around neurons during development, and are specialised forms of neural extracellular matrix (ECM), which have neuroprotective roles and control synaptic plasticity [20,21,89]. Several studies show that, similar to notochordal aggrecan, brain aggrecan does not contain KS; however since most of these studies were conducted in mice and murine cartilage aggrecan does not contain KS, the significance of this statement needs to be carefully evaluated [90,91,92]. Further studies on bovine, ovine and human aggrecan have shown that, while KS is present on brain aggrecan, its content is significantly reduced compared to cartilage aggrecan [90,92,93,94] (Figure 3e). The hydrodynamic size of brain aggrecan is smaller both due to this absence of KS chains and replacement of CS chains with the HNK-1 trisaccharide. Embryonic chick cartilage aggrecan contains KS however notochordal aggrecan does not. HNK-1 aggrecan is also found in early embryonic cartilage rudiments but it disappears with tissue maturation.

The notochord is a full-length embryonic midline structure found in the Chordata [95]. In vertebrates, the notochord is critical for development and defines the major axis of the embryo [96]. The notochord is a source of developmental signals that regulate the patterning of tissues surrounding the notochord [97]. Hedgehog proteins (Shh, Ihh and Dhh) secreted by the notochord are central regulators of embryonic development [98] and control the patterning of tissues and proliferation of cell populations which form a wide variety of organs including the brain, heart and kidneys. Aggrecan interacts with the hedgehog morphogens and has key roles in the regulation of cellular proliferation and tissue development by embryonic NCCs in these tissues. Morphogens orchestrate the actions of progenitor cell populations through the regulation of cellular behaviours including migration, proliferation and matrix deposition into the axial embryonic tissues and in the patterning of the surrounding connective tissues. 

## 7. Co-Ordination of Weight-Bearing and Tension-Bearing Properties in Tissues

Aggrecan equips tissues with an ability to withstand compressive loads and provides mechanical support to elastic and collagenous fibre networks within tissues. These supporting fibre networks provide mainly tensile strength within tissues and are weak in compression. The hydrodynamic space-filling properties conferred by aggrecan therefore allow these tissues to function optimally to resist tensional and shear stresses as well as providing elastic and compressive resilience. Elastic fibrillar structures control reversible tissue deformation providing elasticity to otherwise largely inextensible collagen rich tissues such as cartilage [99,100,101,102]. Historically, the major emphasis of many aggrecan studies were aimed at understanding how aggrecan conveyed functional properties to the weight-bearing articular tissues of diarthrodial joints. The importance of the high fixed charge density of the aggrecan GAG side chains became apparent as an important contributor to the osmotically driven hydration of cartilage which equipped it with the ability to withstand compressive loads [103,104]. However, a few careful studies on aggrecan GAG composition and structure during development, maturation and degeneration also provided important functional information on the GAG side chains of aggrecan. These studies established the importance of GAG sulphation as a functional determinant required not only for aggrecans role in weight bearing but also equip aggrecan with cell directive properties and an ability to interact with morphogens, growth factors and cytokines of importance in tissue development [27,61,105,106].

## 8. Modifications to Aggrecan Side Chain Structure Modifies Its Functional Properties in Tissues 

In adult articular cartilage, aggrecan contains ~100 CS and ~25–30 KS chains, which collectively represent ~90% of the mass of this PG [4]. CS is the predominant GAG in aggrecan and is localised on the C-terminal half of the core protein in so-called CS1 and CS2 domains (Figure 3). KS is also present in a KS rich region between the N-terminal globular domains and the CS rich region. These are *O*-linked through Serine residues to the aggrecan core-protein and have been classified as KS-II chains [2,107]. Complete sequencing of the murine core protein [108,109] shows that it does not contain the consensus sequences for attachment of KS as found in human aggrecan core protein (E-(E,K)-P-F-P-S or E-E-P-(S,F)-P-S) [8,110,111]. Humans and bovine aggrecans contain a 4–23 hexapeptide repeat segment where KS is attached, while rats and other rodents lack this region [110,111]. Rodent aggrecan is truncated in the KS rich region thus does not contain a KS rich region such as that found in human or bovine aggrecan. Rodent aggrecan does however contain small *N*- and *O*- linked KS chains in the G1, G2 and interglobular domain (IGD); IGD KS chains have been proposed to potentiate aggrecanolysis by ADAMTS4 and ADAMTS5 [112]. The lack of a KS rich region in mouse aggrecan does not appear to be detrimental to its normal properties in mouse articular cartilage.

While much still needs to be learnt of the specific roles played by KS in aggrecan, much has already been uncovered about the interactive properties of this GAG in a number of physiological processes in the last decade. Corneal KS-I is interactive with a number of cell stimulatory molecules [113] such as insulin-like growth factor binding protein-2 (IGFBP2) [114], SHH, FGF1 and FGF2 [115,116,117,118]. A proteomics and microarray screen of 8268 proteins and secondary screen of 85 extracellular nerve growth factor epitopes using surface plasmon resonance, micro-array and microsequencing has shown that KS-I interacted with 217 proteins including 75 kinases, membrane and secreted proteins, cytoskeletal proteins, nerve regulatory proteins and nerve receptor proteins [113]. In comparison, chondroitin-4-sulphate interacted with 24 proteins including 10 kinases and 2 cell surface proteins in the same microarrays. Confirmation of these interactions by surface plasmon resonance allowed binding constants to be calculated and the validity of these putative interactions to be determined. Of 85 ECM nerve-related epitopes, KS-I bound 40 proteins, including Slit, two Robos, nine Eph receptors, eight Ephrins, eight Semaphorins and two nerve growth factor receptors. It has yet to be ascertained however if the KS-II chains of aggrecan have similar interactive properties as KS-I. 

Antibodies which detect low sulphation KS motifs have now been developed (reviewed in [119]) and have demonstrated KS in a number of tissues previously thought to be KS deficient after labelling with mAbs such as 5-D-4, which is specific for highly sulphated KS epitopes [120]. Roles are emerging for low sulphation KS-epitopes in electro-sensory processes [69,115,121]. Neural tissues are the second richest source of KS in the human body after the cornea [69,115].

While aggrecan has important interactions with growth factors and morphogens which direct chondrocyte proliferation and differentiation in cartilage development and maturational processes essential in endochondral ossification and skeletogenesis, it also has important functional roles to play in weight-bearing and in the stabilisation of the cartilage ECM. Aggrecan, as its name indicates, forms massive mega Dalton aggregate ternary complexes via interaction of its N-terminal HA binding G1 domain with hyaluronan (HA) stabilised by cartilage link protein which shares homology with the G1 domain and also has HA binding properties [2,4,107]. The G3 domain of aggrecan also interacts with tenascin-C via its fibronectin type III repeats, which have lectin binding activity, and these interact with the C-type lectin motifs on the aggrecan G3 domain [19,122,123,124]. Tenascin-C, R, Fibulin-1 and fibulin-2 also bind to the cartilage aggrecan G3 domain through interactions with its C-type lectin and EGF domains of G3 [15]. The C-type lectin of the aggrecan G3 domain also interacts with cells and activates the Complement system [124]. Complement is a defence system against foreign pathogens and aids in the removal of dying cells, immune-complexes, misfolded proteins and invading microbes [125]. Excessive complement activation can exacerbate autoimmune disorders and pathological inflammatory conditions such as rheumatoid arthritis (RA) [126]. Complexes of matrilin-1 and -3 and biglycan or decorin also connect collagen VI microfibrils to collagen II and aggrecan [127], forming a link between the PG and fibrillar collagenous networks in cartilage and IVD [1,3,5]. Cartilage oligomeric protein (COMP and TSP-5) also binds to aggrecan, providing an extended co-operative network in cartilage [128], which helps to distribute loading stresses throughout this tissue avoiding the point loading which can be damaging to ECM components [3]. This extended collagen–aggrecan network also provides a mechanosensory biosensor system extending far from the cell through the interstitial and inter-territorial matrix, which allows the chondrocyte to perceive and respond to perturbations not only in its local mechanical microenvironment but also to more remote cartilage regions to regulate tissue homeostasis and optimal tissue functional properties [1,3].

The essential role of aggrecan to cartilage function is well illustrated in a naturally occurring Cmd (cartilage matrix deficient) mutant mouse [129], which has a single 7 bp deletion in exon 5 of the aggrecan gene which encodes the B loop of the G1 domain of aggrecan [130]. Homozygote (cmd/cmd) mice display dwarf-like features, spinal deformity, chondrodysplasia, abnormal collagen fibrillogenesis, a cleft palate [130], deafness [131] and die shortly after birth due to respiratory failure [132]. The articular cartilage of the Cmd^−^/Cmd^−^ mouse displays tightly packed chondrocytes surrounded by little matrix; growth plate cartilage contains chondrocytes arranged in disorganised columns of diminished length in severely diminished proliferative, pre-hypertrophic zones consistent with the reduced proportions of these mice [132] (Figure 4a). Cultured nanomelic chick chondrocytes synthesise a truncated aggrecan core protein precursor [133] due to a premature stop codon, and this is not translocated to the Golgi apparatus for processing, which leads to an absence of aggrecan in nanomelic cartilage, chondrodysplasia, disrupted organisation of the hyaline and growth plate cartilages, severely diminishing skeletal stature [134,135,136,137] (Figure 4b). 

While the role of the KS chains in the G1 and G2 domains of aggrecan is largely unknown, some G1 KS chains have been found to sterically obscure an N-terminal T cell attachment site in aggrecan and have a protective effect over autoinflammatory conditions arising from fragmentation of aggrecan (Figure 5a,b). Further T cell interactive sites in the G3 domain of aggrecan have also been identified which may contribute to auto-inflammatory arthritic conditions [138,139,140]. These G1 KS chains suppress a T cell mediated response initiated by free G1 when it is used as an arthritogen in models of inflammatory arthritis [138,139,140,141,142]. KS chains in the IGD also potentiate aggrecanase activity in this region of the core protein [143]. A few KS chains are also interspersed within the CS rich region. KS-II chains in aggrecan from weight-bearing tissues such as articular cartilage and IVD contain 1-3 fucose and 2-6 *N*-acetyl-neuraminic acid residues [119]; however, these modifications in KS are absent in aggrecan from non-weight-bearing nasal and tracheal cartilage [144]. The significance of these KS modifications and why they only occur in aggrecan from weight-bearing tissues is unknown; antibody 3D12/H7 identifies these KS chains embedded in the CS rich region [145] but they do not share immunological identify with KS chains in the KS rich region. This KS epitope contains three sulphate groups and two fucose residues on GlcNAc residues in a branched fucosylated sialo-KS structure of unknown function. 

Several mutations in the aggrecan gene have been documented, which affect variable regions in the aggrecan core protein leading to a number of conditions collectively termed the aggrecanopathies (Figure 5c) [146,147]. The aggrecanopathies are a spectrum of non-lethal skeletal dysplasias including spondyloepimetaphyseal (SEMD) and spondyloepiphyseal dysplasia (SED), osteochondritis dissecans (OCD) and a number of accelerated bone maturation disorders that result in short stature and idiopathic short stature (ISS) [146,148]. Skeletal abnormalities are also prominent features of animal models which display deficient levels of cartilage aggrecan such as the Cmd mouse [129,130] or nanomelic chick [137,149]. Brachymorphism [150] also results in reduced PAPS levels, and the aggrecan synthesised is deficiently sulphated and functionally impaired, resulting in abnormal skull development and short squat skeletal frames [151,152,153]. Manipulation of the diastrophic dysplasia sulphate-transporter gene (*DDST*) also results in the synthesis of aggrecan with deficient sulphation levels and a variety of skeletal abnormalities such as short stature and joint dysplasia in diastrophic dysplasia [151], micromelia in atelosteogenesis Type II [152] and short skeletal proportions due to aberrant trunk and extremity development in achondrogenesis Type II. Heterozygous ACAN mutations result in a phenotypic spectrum of skeletal abnormalities including short stature, accelerated bone maturation, early growth cessation, poor responsiveness to human growth hormone, brachydactyly, early-onset OA and susceptibility to the development of degenerative disc disease due to dysfunctional articular cartilage and IVD tissues [147,154,155,156]. Osteochondritis dissecans (OCD) is a disabling condition characterised by abnormal deposition of aggrecan in cartilage and the appearance of cracks in the cartilage and subchondral bone. This condition effects juveniles and adults but its aetiology is unknown. Trauma has been suggested as a predisposing factor in juveniles and recent genomic wide studies have identified a cluster of genes associated with this condition suggesting that it may also have a genetic basis [157,158,159]. Several skeletal dysplasias have been shown to be due to a constitutively activated mutation in a transient receptor potential vanilloid 4 (TRPV4) cation channel protein [160,161,162,163,164]. This results in abnormal cation mediated cell signalling by chondrocytes and altered regulation by BMP2 and TGFβ1 activity [164].

Aggrecan is required for correct growth plate cytoarchitecture and differentiation, endochondral ossification and skeletogenesis [165]. The CS side chains of aggrecan make important contributions to this process and their sulphation status is an important functional determinant [60]. Six CS *N*-acetylgalactosaminyltransferases (t1–t6) have been described. Initial stages of CS sulphation is undertaken by t1 and t2; t1 and t2 double knockout mice display shortened growth plates, distorted hypertrophic growth plate regions, reduced growth plate chondrocyte proliferation, type X expression, dwarfism, disruption in the postnatal formation of the secondary ossification centres and chondrodysplasia; this is lethal postnatally [166] (Figure 6). Aggrecan aggregates are also formed in the CNS and PNS stabilised by interaction with tenascin-C and tenascin-R, HA and a brain specific link protein variant Bral-1 (HAPLN2) to form perineuronal nets (PNNs), which are structures assembled around neurons (Figure 7) that scavenge oxygen free radicals in neural tissues thus preventing oxidative stress [19,20,93,167,168,169]. Brain tissue is fatty acid rich and prone to oxidative damage, which produces reactive species that detrimentally affect mitochondrial activity in neurons. Brain tissue is metabolically demanding and requires optimal mitochondrial activity to ensure energy production to power neuronal signalling.

## 9. Aggrecan–GAG Interactions Are of Importance in Heart Development 

During early embryonic development, ectodermal NC cells migrate to form the neural tube and notochord under the direction of HNK-1 substituted aggrecan. This HNK-1 substituted form localises to the peri-notochordal space where its repulsive cell interactive properties guide the NC cells to form the notochord in a co-ordinated pattern-dependant manner (Figure 8). Equally impressive is the direction of assembly of tissue structures through HNK-1 aggrecan with NC cells migrating outwards along specific guidance tracts to form the cardiac neural crest region and the cardiac septae, outflow tracts and aortic arches [171] (Figure 9). Development of the heart valves and cardiac muscle with electroconductive properties from the endocardial cushions is also regulated by distinct spatiotemporal distributions of aggrecan and versican [16,172]. Heart tissues have remarkable mechanical properties of elasticity, compressibility, stiffness, strength and durability achieved through careful guidance of cell-mediated ECM assembly of collagen fibril and versican- and aggrecan-rich tissue regions to provide these tissues with highly specialised functional properties. Cardiac tissues are electroconductive and the charge transfer properties of GAG side chains of heart PGs may contribute to tissue properties in a similar manner to how electrosensory properties are conveyed to neural tissues. Similar developmental processes are also evident in the formation of cartilage, tendon and bone using the same ECM components but in a different manner to effect specialised tissue function [11]. The development of co-ordinated electroconductive cardiomyocyte networks with synchronised pulsatile properties is a particularly impressive achievement [173,174]. The properties of the heart valves and heart strings are equally important to heart function and these have material properties more similar to cartilage and tendon. It is not surprising therefore that aggrecan and transcription factors such as Sox 2, Sox 9 and growth factors/morphogens such as FGFs/BMPs play such prominent roles in the development of cardiac tissues [11]. 

The calcineurin/nuclear factor of activated T cells (NFatc1), which regulates osteoclast differentiation [180], is also required for valve formation [181,182]. Myocyte-specific enhancer factor 2C (Mef2c), a master transcription factor which regulates hypertrophy and osteogenic differentiation of chondrocytes [183], is also essential for normal cardiovascular development, and loss of function mutations in Mef2c contribute to congenital heart defects [184]. Moreover, activated BMP signalling has been shown to increase expression of cartilage and bone-type collagens, and increased expression of the osteogenic marker Runt-related transcription factor 2 (Runx2)/core-binding factor subunit alpha-1 (CBF α-1) is observed in adult aortic valve disease [185]. Thus, there is considerable overlap in cartilage synonymous transcriptional factors in the integrated development of functional heart tissues. The cell directive properties of HNK-1 aggrecan not only makes a particularly important contribution to the sculpting of embryonic cardiac tissues but it is also a functional component of these tissues.

## 10. Aggrecan and Cellular Regulation

The form of aggrecan present in the notochord does not contain KS but contains the HNK-1 trisaccharide recognition motif [92] (Figure 3b,e). S103L reactive aggrecan is prominent in the peri-notochordal space and its inhibitory properties on NC cells instructively guides their migration during formation of the neural tube and notochord [90,91] (Figure 9). The absence of KS on aggrecan is not without precedent. Rodent aggrecan has a truncated core protein and also does not contain a KS rich region; however, this is not detrimental to its weight-bearing properties in cartilage or the turnover of aggrecan by MMPs in these tissues. Rodent aggrecan contains small *O*- and *N*-linked KS chains in the G1, IGD and G2 domain described in human aggrecan with roles in the potentiation of ADAMTS-4 and -5 activity. Some studies have also reported the existence of populations of aggrecan devoid of KS in brain tissues based on an absence of reactivity with 5D4 anti-KS antibody, although their detailed characterisations have yet to be provided. Aggrecan expressed by embryonic glial cells in the brain is an astrocyte developmental regulator [186]. Chick aggrecan nanomelia mutants display marked increases in the expression of astrocyte differentiation genes in the absence of extracellular aggrecan indicating that aggrecan regulates astrocyte differentiation and controls glial cell maturation during brain development [186]. Heavily sulphated CS chains on aggrecan and other PGs can bind the midkine family members (midkine and pleiotrophin), and some FGF family members (FGF-1, -2, -16 and -18) provide clues as to how they influence cellular processes [48,187]. Appican in brain tissue [188], a CS-PG synthesised by astrocytes but not by neurons [189], contains embedded CS-E motifs [190] within their CS side chains which interact with heparin-binding neuroregulatory factors [187]. Expression of Appican by astrocytes induces morphological changes in C6 glioma cells and promotes adhesion of neural cells to the ECM [191].

In articular cartilage, the CS chains of aggrecan have major roles in the attraction of water into this tissue which forms the basis of its hydrodynamic viscoelastic properties as a weight-bearing tissue. However the non-reducing terminal regions of the CS chains of aggrecan from articular and growth plate cartilages also contain 4, 6-disulphated CS, and these are likely binding candidates for morphogenetic proteins which show a similar distribution to aggrecan in these tissues [192]. BMP-2, FGFR-3 and IHH co-distribute in growth plate cartilage with the pre-hypertrophic cells [193]. BMP-2, BMP-4 and dual BMP-2/4 knockout mice have severely disturbed shortened growth plate organisation due to decreased chondrocyte proliferation and increased apoptosis [194,195,196]. Type X collagen expression is also severely down regulated as is MMP-13 expression in BMP-2/4 KO mice.

## 11. Role of IHH in Chondrogenesis

Indian hedgehog (Ihh) [197], a member of the hedgehog protein family along with sonic hedgehog (Shh) [198], regulates chondrocyte differentiation, proliferation and maturation in articular cartilage development [199] and during endochondral ossification through interactions with parathyroid hormone-related peptide (PTHrP) [200] and BMP mediated cell signalling [201]. Ihh has multiple functions during skeletogenesis [202,203,204]. Mice lacking the Ihh gene exhibit severe skeletal abnormalities, including markedly reduced chondrocyte proliferation and abnormal maturation and an absence of mature osteoblasts, which has detrimental effects on bone development [205]. Ihh and its receptor, smoothened (smo), are expressed in chondrocytes and osteoblasts thus Ihh may have a direct effect on osteoblasts, or its effects may be mediated indirectly through chondrocytes during the process of endochondral ossification.

IHH colocalises with aggrecan in the growth plate (Figure 4A,B; plate A g; B c,f). Aggrecan regulates the expression of growth factors and signalling molecules during cartilage development and is essential for proper chondrocyte organisation, morphology and survival during formation of the axial skeleton. The sulphated GAGs of the CS and KS side chains of aggrecan provide water imbibing properties creating a large hydrophilic molecule important for the hydration of cartilage and the provision of its hydrodynamic weight-bearing properties but also bind growth factors and morphogens crucial to chondrocyte maturation and function [27,206]. Thus, aggrecan should not be considered merely as a space-filling ECM component that provides hydration and weight-bearing properties to tissues but also a cell directive tissue organiser that is capable of modulating the activity of growth factors and morphogenetic proteins, thus mediating tissue development. Indeed, aggrecan knock-out mutants display a range of severe ECM defects, which supports this proposal [134,165].

## 12. HNK-1 Carbohydrate Epitope as a Recognition Motif

The human natural killer-1 (HNK-1) carbohydrate motif is a unique sulphated trisaccharide of the structure SO_4_-3GlcAβ1-3Galβ1-4GlcNAc, which is developmentally and spatially expressed in a cell-type specific manner within the CNS (Figure 3a–d) [207]. HNK-1 is also a well-known CNS glycoprotein epitope with essential roles to play in neural plasticity, higher brain function, synaptic plasticity, spatial learning and memory. However, it is not limited to the CNS and also displays specific localisations in other strategic locations elsewhere in the human body including in the kidney [208], the heart [209,210], retina [211] and as a component of the PNNs identified in the auditory system [212]. As a neural cell marker, HNK-1 plays crucial roles in cell migration and cellular attachment during embryonic nerve development and formation of the notochord from the neural tube (Figure 8). Carbohydrate–protein interactions between HNK-1 reactive sulpho-glucuronyl-glycolipids and PG lectin domains mediate neuronal cell adhesion and neurite outgrowth [213,214,215]. Some laminin isoforms also bind specifically to sulphated glycolipids [216] and are important in cell adhesion [215], particularly in nerve development. Several neural cell-adhesion molecules contain the HNK-1 epitope including neural cell adhesion molecule (NCAM), myelin associated glycoprotein (MAG), myelin basic protein (MBP), neural-glial adhesion molecule (Ng- CAM, L1), contactin, P0, Tenascin-C, Tenascin-R. Sub-populations of the enzymes *N*-acetylcholinesterase and 5′-nucleotidease also contain the NHK-1 epitope, which is important in their localisation in synaptic vesicles and membranes. Acetyl cholinesterase and 5′-nucleotidase are GPI anchored ecto-enzymes of high catalytic efficiency. Acetylcholinesterase cleaves the neurotransmitter acetylcholine in the neuromuscular junction (NMJ) and this allows muscles to return to a relaxed state following contraction. Acetylcholinesterase cleaves in excess of 5000 molecules of acetylcholine/s per molecule of enzyme. Without such an efficient enzymatic system and co-ordinated expression of neurotransmitters in the NMJ, muscles would be tensed and relaxed in an uncoordinated manner and movements would be jerky and irregular as evident in neuromuscular disorders such as Parkinson’s disease or the spastic paralysis evident in Schwartz–Jampel Syndrome. It is not surprising therefore that the expression of NHK-1 is under strict spatial and temporal regulation on migrating neural crest cells, cerebellum, myelinating Schwann cells and motor neurons but not on sensory neurons. A form of HNK-1 substituted aggrecan is synthesised in the notochord and in early foetal rudiment cartilage [90,92] (Figure 9a,b). HNK-1 has been immunolocalised to the electro-receptors of the shark and electric organs of the electric eel (*Electrophorus electricus*), electric catfish (*Malapterurus electricus)* and electric ray (*Torpedo marmorata)*. HNK-1 has also been mapped to electroconductive tissue during human foetal heart development and is expressed by cultured cardiomyocytes leading to the development of smart electroconductive polymers for applications in regenerative medicine [217]. HNK-1 sulphotransferase (HNK-1ST) catalyses the transfer of sulphate to position 3 of terminal glucuronic acid in protein and lipid linked oligosaccharides carried by many neural recognition molecules [218,219]. These facilitate cellular interactions during CNS development and in synaptic plasticity. HNK-1 ST acts in combination with two other glucuronyl transferases (GlcAT-P and GlcAT-S) to form a heteromeric complex in the biosynthesis of the HNK-1 trisaccharide epitope [220,221]. HNK-1 ST suppresses the glycosylation of α-dystroglycan in sub-populations of melanoma cells in a number of tissues where neither GlcAT-P nor GlcAT-S is expressed, and this reduces the ligand binding capability of α-dystroglycan establishing a tumour suppressor role for HNK-1 ST in melanoma. The HNK-1 epitopes of acetylcholinesterase and 5′-nucleotidase have roles in cell–cell and cell–matrix communication independent of their enzymatic activities.

HNK-1 (SO_4_-3GlcAβ1-3Galβ1-4GlcNAc) is expressed on *N*-linked and *O*-mannose linked glycans in the nervous system. Several proteoglycans bear the HNK-1 epitope including phosphacan and aggrecan (Figure 3b,c). NHK-1 sulphotransferase can utilise the xylose-galactose-galactose-glucuronic acid linkage tetrasaccharide as acceptor to attach the 3-*O* sulphate group to the non-reducing glucuronic acid residue [222] but in so doing inhibits further CS chain elongation; thus, aggrecan substituted with HNK-1 has a lower density of CS chains and has a reduced hydrodynamic size compared to cartilage aggrecan [220,222]. Phosphacan occurs as a soluble PG and as a variant protein tyrosine phosphatase which contains KS and CS side chains in addition to HNK-1 carbohydrate. The HNK-1 motif in phosphacan is *O*-mannose linked through an Asn on the core protein. Notochordal and early rudiment cartilage cells synthesise a form of aggrecan substituted with the HNK-1 epitope, but this disappears in later stages of skeletal development (Figure 8 and Figure 9).

The HNK-1 substituted Tenascin-R and -C splice variant multimeric ECM glycoproteins contain multiple FNIII and EGF repeats and a fibrinogen domain which are interactive with the C type lectin domains of the lectican CS PG family in brain [223,224]. Tenascin-R is a major component of the PNNs which surround neurons in the brain, spinal cord and in specific areas of the hippocampus [225]. Perineuronal nets consist of the lectican CS-PGs assembled into extracellular networks through interaction with HA and link proteins cross-linked by Tenascin-R and are linked to neurons through their C-terminal domains, endowing them with neuroprotective properties [226] (Figure 7a–e,k).

## 13. The Therapeutic Potential of Aggrecan and Its GAG Side Chain Components

### 13.1. Analysis of Cartilage Aggrecan and Its GAG Side Chains

As already shown in this review, aggrecan is a large specialised protein which provides weight-bearing or space-filling properties to cartilaginous tissues through its large solvation volume and ability to imbibe water. Cartilage aggrecan has a core protein of ~250 kDa and contains ~100 CS and 25–30 KS side chains, which collectively represent ~90% of its mass. As also shown in this review, aggrecan forms also exist in specialised tissue niches and in developmental contexts which do not contain KS or have some CS chains replaced by the HNK-1 trisaccharide which result in changes in aggrecan’s interactive properties. Murine aggrecan contains a truncated core protein and is devoid of a KS rich region; however, this does not impede its functional properties in murine cartilage or the normal turn-over of this proteoglycan. Thus, the functional importance of KS in human aggrecan is unknown at present and the need for two GAG types in aggrecan is a question which has yet to be answered. While corneal KS-I has interactive properties with a range of neuron associated proteins [113,119] such as SHH, FGF-1 and FGF-2 [117], it is not known if the KS-II chains of aggrecan share this property. KS-II differs from KS-I in its capping modifications with l-fucose and *N*-acetyl neuraminic acid [119], which render KS-II resistant to total depolymerisation by keratanase-I and II and endo-β-d-galactosidase apparently through steric constraints which prevent access of these KS depolymerising enzymes to KS substrate, KS-I is totally depolymerised under the same digestion conditions thus significant differences exist between KS-I and KS-II [227]. This could also sterically impede potential interactions of KS-II with other ligands. Furthermore, cartilage aggrecan also contains a few KS chains interspersed within its CS-1 and CS-2 rich regions. An antibody to these KS chains, (MAb 3D12/H7) identifies trisulphated fucosylated and poly-*N*-acetyllactosamine modifications in the KS linkage regions in these KS chains to aggrecan core protein [145]. These 3D12/H7 positive KS chains do not share immunological identity with the KS-II chains of the KS rich region of aggrecan however their functional properties still have to be ascertained.

To understand the properties of the aggrecan side chain GAGs and how these may contribute to the properties of aggrecan, methods have been developed to isolate aggrecan and its GAG side chains from cartilaginous tissues.

### 13.2. Aggrecan Isolation Procedures

To isolate aggrecan from cartilage for analysis, it must be dissociated from its ternary complex formation with HA and link protein. This is achieved by using chaotropic agents such as guanidinium hydrochloride (GuHCl), which disrupts the water structure of the tissue, opens up the dense collagenous structure allowing dissociation of the aggrecan–HA–link protein complexes and release of aggrecan monomer which diffuses out of the tissue and is recovered in the extraction solution [228]. Cartilage is initially diced into small pieces to reduce the diffusive pathways out of the tissue for effective extraction; broad spectrum protease inhibitors covering all four mechanistic classes of proteases are included in the extraction solution to protect the aggrecan from proteolysis. Homogenisation procedures, which are commonly used in the extraction of proteins from other soft connective tissues, cannot be used for the isolation of aggrecan in an intact form since their high shear forces fragment the aggrecan. The cartilage extract can then be subjected to anion exchange chromatography on support matrices derivatised with anionic ligands such as diethylaminoethyl (DEAE) or sulphopropyl [229], dissociative size exclusion chromatography in 4-M GuHCl containing buffers using open pore gel chromatographic media such as Sephacryl or Sepharose CL2B [230], or density gradient equilibrium isopycnic ultracentrifugation in high concentrations of CsCl [231]. This latter procedure relies on aggrecan’s high buoyant density in CsCl gradients of ≥1.55 g/mL to isolate aggrecan; while extracted proteins typically have buoyant densities of 1.3–1.35 g/mL, HA has a buoyant density of 1.4–1.45 g/mL. Density gradient ultracentrifugation can be conducted in the presence of 4M GuHCl to ensure aggrecan is isolated free of other interactive components also present in the cartilage extract [231,232].

### 13.3. Analysis of Aggrecan’s GAG Side Chains

The CS chains of aggrecan are primarily of interest since these make a major contribution to aggrecans physicochemical and biological properties in tissues, whereas the function of the more minor KS chains are currently not known and thus are of lesser interest [116,233,234,235]. Several qualitative and quantitative methods have been developed for the discriminative measurement of intact GAG chains, including dye specific, thin layer chromatography (TLC), capillary electrophoresis [236,237,238], high-performance liquid chromatography (HPLC), various mass-spectrophotometric formats including liquid chromatography–tandem mass spectrometry (LC-MS/MS) [239,240,241], gas chromatography, enzyme linked immunosorption analysis (ELISA) using a wide array of anti-GAG antibodies, GAG microarrays and automated high-throughput mass spectrometric methods. Electrophoretic methods to separate intact GAG chains and GAG disaccharides generated by GAG depolymerising enzymes by capillary electrophoresis and conventional slab gel formats use media such as highly purified agaroses [242], polyacrylamide or mixtures of these as separation media. GAGs can be descriminated enzymatically either by eliminative cleavage with lyases (EC 4.2.2.-) or by hydrolytic cleavage with hydrolases (EC 3.2.1.-). These enzymes can be used in combination with chromatographic or electrophoretic separation methodologies to identify GAG species [243,244,245,246,247]. Following electrophoresis, electroblotting of the separated GAGs can be employed to nylon or nitrocellulose support membranes treated with cationic detergents. A large array of specific anti-GAG antibodies can be used for identification of the blotted GAGs. In gel, detection of separated GAG disaccharides and oligosaccharide species can also be carried out using fluorophore assisted carbohydrate electrophoresis (FACE) [248]. Capillary electrophoresis, has high resolving power and sensitivity in the analysis of GAG composition, disaccharide sulphation patterns and sequence analysis [236,237,238].

## 14. Evaluation of the Aggrecan Content and Distribution in Pathological Cartilage Using Imaging Techniques

This review shows that aggrecan is an important functional component of articular cartilage and is depleted in OA cartilage. Several non-invasive cartilage imaging procedures have been developed that allow the assessment of the spatiotemporal distribution of aggrecan during OA disease progression. MRI of articular cartilage (AC) has been applied to assess osteoarthritic changes occurring in cartilage with the progression of OA. Traditional MRI evaluates AC morphology and measures cartilage thickness over time [249]. More advanced MRI techniques can now be used to assess AC matrix composition non-invasively to detect early articular changes. T2-mapping and T1ρ sequences estimate the relaxation times of water inside AC and have found application in clinical protocols to assess cartilage changes in OA [250]. Diffusion-weighted and diffusion tensor imaging can also be used to assess ECM changes in AC since the movement of water in cartilage is affected by ECM composition and structure. Specific imaging techniques that evaluate cartilage GAGs, such as delayed gadolinium enhanced MRI [251] or Chemical Exchange Saturation Transfer [252,253] and sodium imaging [254,255] or PET (positron emission tomography)-sodium imaging [256] have also shown utility in the non-invasive detection of AC damage [257].

## 15. Tissue Therapeutic Interventions Involving CS

With the recent publication of the first draft of the GAG Interactome, the GAG–protein interactive properties previously investigated [116,233] have now been extensively catalogued [235]. There are two major areas of therapeutic application involving CS: (i) use of CS as a drug to treat OA cartilage depleted of proteoglycans; and (ii) therapeutic use of CS-depolymerising enzymes to remove the CS side chains of CS-PGs that are laid down in scar tissues, which stabilise spinal cord defects, neural damage in the PNS and neural damage following brain trauma [243,258,259]. While CS-PGs are laid down in neural scar tissues to stabilise the neural defects to prevent further mechanical damage at the defect site, the CS side chains of these PGs inhibit neural outgrowth through the scar tissue, thus functional recovery of the spinal cord or other traumatised neural tissues is prevented [260]. Chondroitinase ABC, ACII and hyaluronidase-4 (HYAL4) [261], which is a CS hydrolase despite its misnaming, have all been evaluated in models of spinal cord injury [262]. Removal of CS from the defect site by these enzymes resulted in recovery of neural functional properties [260,263,264,265]. Acute trauma to the brain and upper limbs resulting in neural damage have also been treated using chondroitinase ABC [266,267], resulting in neural sprouting through the defect site [268] and functional recovery [269].

Chondroitin sulphate has been of interest as a therapeutic agent for the treatment of OA for at least the last decade. A biochemical study in 2008 showed that CS interfered with progressive degenerative structural changes in joint tissues and thus showed promise in the treatment of OA [270]. A review of several CS preparations subsequently shows variable but generally positive responses in the treatment of OA but emphasises the need for highly purified CS preparations to provide unequivocal results [271]. A further study with pharmaceutical grade CS subsequently confirmed that highly purified CS had beneficial effects in the treatment of OA [272]. Continued assessment of highly purified pharmaceutical grade CS and other CS preparations confirmed safety data on the use of pharmaceutical grade CS for the treatment of OA and set down some guidelines for its use but indicated these were not applicable to lower grade CS preparations [273]. A further study raised doubts on some therapeutic CS preparations primarily focussing on the beneficial functions of CS-based therapeutic supplements and potential harmful effects of some fucosylated CS preparations which may contaminate these in a similar manner to the oversulphated CS species which have previously been identified, as contaminants in some heparin preparations [274]. A current study on highly purified therapeutic commercially available CS (Condrosulf^®^, IBSA, Biochimique, Lugano, Switzerland) and a literature review on its clinical efficacy confirmed the reduced pain and improvements in joint function afforded by this preparation of CS to OA patients [275]. Condrosulf^®^ is a cost-effective and safe treatment for OA, efficacious after 30 days of administration and has beneficial properties for at least the months after the drug is discontinued. Full safety reports analyses confirmed the safety profile for CS. It has almost no side effects and shows better gastrointestinal tolerance compared to conventional non-steroidal anti-inflammatory drugs used to treat OA [275]. The beneficial properties of CS may explain the resurgence in the use of PGs, recombinant PG sub-domains, GAG and PG mimetic, and the development of neo-PGs [276] for therapeutic repair procedures. CS can also be used to stimulate stem cells and promote the attainment of defined pluripotent stem cell lineages [277]. GAGs also have generalised properties which are useful in tissue repair [278] and have been incorporated into a number of bioscaffolds to promote stem cell regulation and to develop potential new tissue repair strategies (reviewed in [279]).

## 16. Conclusions

*N*-terminal interactions of aggrecan with HA forms macro-aggregate structures that are physically entrapped within the type II collagen networks of cartilaginous tissues. These along with the C-terminal G3 mediated interactions of aggrecan EGF-like, Complement-like and C-type lectin domains with Type VI collagen lattices forms an extended tissue-wide cooperative network. This network provides a bio-sensory platform that not only dissipates loading in cartilaginous tissues but also facilitates an appropriate metabolic response by chondrocytes within the tissue to the loads they perceive to assemble an optimal functional and protective matrix, thus preventing cellular damage through point loading. Matrilin-3, SLRPs, COMP and fibulin also stabilise G3 interactions in this network. Collectively, these N- and C-terminal interactions tether aggrecan at both ends in tissues where its dynamic water imbibing properties with HA convey weight-bearing properties to tissues. In cartilaginous tissues, incorporation of aggrecan into an extended mechano-transductive sensory network allows the resident chondrocytes to perceive and respond to biomechanical changes to maintain biosynthetic responses which ensure tissue homeostasis and optimal tissue properties. The interactive properties of the G1 domain of aggrecan with HA undergoes a maturational phase where initially it does not interact with HA for 24 h which allows the newly secreted aggrecan to diffuse away from the cell into the interstitial matrix, as confirmed through pulse-chase radiosulphate labelling experiments. Immunohistochemistry also demonstrates a high density of aggrecan in the pericellular matrix around chondrocytes where network formation transmits regulatory cues to the chondrocyte to effect tissue homeostasis and aid in the stabilisation of cartilage. The variable functional attributes of aggrecan in particular tissue contexts is due to the diverse structure of its attached glycan side chains, allowing it to act as a space-filling molecule with an ability to entrap water in weight-bearing tissues such as articular cartilage and IVD but also as an interactive molecule with morphogens, growth factors and cells in growth plate cartilage and embryonic tissues. This functional diversity arises through substitution and post-translational modifications of aggrecan’s attached GAG chains, for example post-translational modification of CS can occur such as disulphation on some non-reducing termini or variation in the disaccharide composition along the CS chain. These disulphated terminal groups regulate collagen fibrillogenesis in growth plate cartilage and are interactive with morphogenetic proteins such as IHH and SHH. These morphogenetic proteins direct maturational changes in the growth plates by regulating spatial and temporal chondrocyte differentiation eventually leading to growth plate closure and mineralisation at the cartilage–bone interface as part of the endochondral ossification process to extend the axial skeleton. The molecular composition of aggrecan thus varies in specific cellular and developmental contexts. Aggrecan in early embryonic development of the neural tube and notochord does not contain KS chains and some of its CS chains are replaced by the HNK-1 trisaccharide motif. This reduces aggrecans charge density and its solvation volume but conveys homophilic and heterophilic HNK-1 mediated properties and interactions with NC cells and ECM glycoproteins which direct NC cell migration, development of the neural tube and notochord and migration of precursor cells involved in the development of the neural network, heart and brain stem. Aggrecan also forms specialised ECM structures such as PNNs in the CNS/PNS, which are neuroprotective and important for synaptic plasticity and cognitive learning. The form of aggrecan in these structures contains KS, CS and the HNK-1 glycan motif; however, the density of attached CS and KS side chains in brain aggrecan is less than in cartilage aggrecan. While aggrecan has essential cell directive properties in embryonic heart formation, it also has important supportive roles to play in mature heart tissues. The heart has a complex structure, aggrecan is an important functional component of a number of its tissues providing mechanical strength and resilience for the demanding continuous cycles of compression and relaxation which occur throughout an animals life-time. Aggrecan also stabilises the attachment points of valve leaflets facilitating the co-ordinated flow of blood between the ventricles and strengthens the heart tendon chordae tendineae which attach the papillary muscles of the internal heart wall to the atrioventricular valve. These are important internal stabilising structural components of the heart. This review shows that aggrecan has a diverse range of functional attributes and is of major importance not only in embryonic skeletal development but also in mature tissues where it maintains homeostasis and functionality. Due recognition of aggrecans attached GAG chains is important and explains its diverse tissue context driven properties. A greater understanding of the glyco-code and its cell directive properties may one day provide important insights as to how specific tissue repair and regenerative strategies may be directed more effectively in repair biology.

## Figures and Tables

**Figure 1 biomolecules-10-01244-f001:**
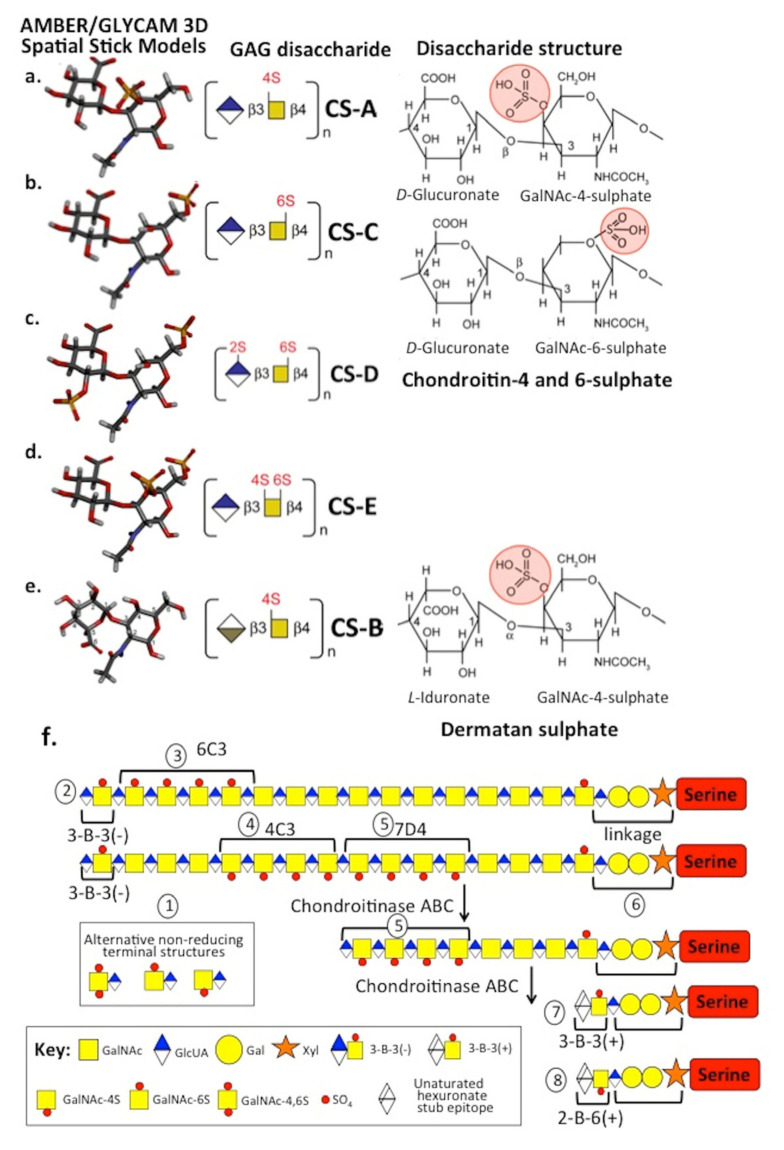
Amber/GLYCAM 3D stick structures of chondroitin sulphate isomers depicting their 3D conformations, disaccharide compositions and Haworth projection disaccharide structures showing sulphation positions (**a**–**e**). Schematic depiction of the structural organisation of the chondroitin sulphate glycosaminoglycan side chains of aggrecan depicting specific structural features of areas of the chain detected by monoclonal antibodies, putative sulphation patterns, linkage region structure to aggrecan core protein and non-reducing terminal structures (**f**). These regions on the CS side chain are numbered 1–8. Key: (1) Non-reducing terminal groups present on some cartilage aggrecan CS chains; (2) 3-B-3(−) CS sulphation motif is also present as a non-reducing terminal component on some chains; (3) putative region on CS chain identified by MAb 6C3; (4) putative region on CS chain identified by MAb 4C3; (5) putative region on CS chain identified by MAb 7D4; (6) CS linkage attachment region to Serine residues of the aggrecan core protein; (7) 3-B-3(+) CS sulphation stub epitope generated by exhaustive digestion of the CS chain by chondroitinase ABC and recognised by MAb 3-B-3; and (8) 2-B-6(+) CS sulphation stub epitope generated by exhaustive digestion of the CS chain by chondroitinase ABC and recognised by MAb 2-B-6. Note: Regions 3–5 of the CS chains containing the 6-C-3, 4-C-3 and 7-D-4 reactivity are susceptible to chondroitinase ABC digestion; thus, in graded partial digestions, the 6-C-3 and 4-C-3 reactivity can be selectively removed leaving the 7-D-4 reactive region intact. However, this is also susceptible to chondroitinase ABC, and exhaustive digestion conditions eventually lead to generation of the unsaturated 3-B-3(+) and 2-B-6(+) stub epitopes attached to the linkage region, as shown in this diagram. In (**f**), the structures shown hypothetical many features such as the sulphation positions on GAGs are variable; the depictions shown are thus generalisations based on literature data.

**Figure 2 biomolecules-10-01244-f002:**
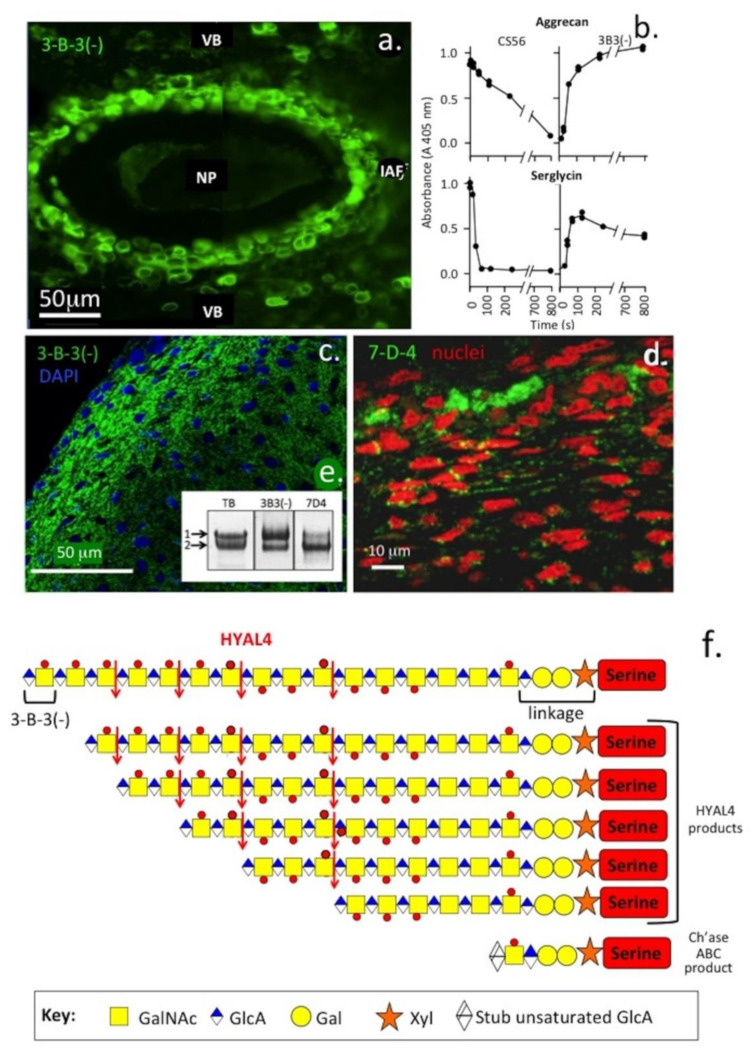
Immunofluorescent localisation of the 3-B-3(−) CS sulphation motifs on aggrecan associated with rudiment cartilage of a rat intervertebral disc (**a**) and demonstration of the generation of the 3-B-3(−) epitope by digestion of aggrecan and serglycin as model proteoglycans with hyaluronidase-4 (HYAL4) (**b**). Immunolocalisation of the 3-B-3(−) and 7-D-4 CS sulphation motifs in developmental human foetal knee joint cartilage (14 weeks gestational age) (**c**,**d**). The inset of (**e**) shows foetal aggrecan samples separated by native composite agarose polyacrylamide gel electrophoresis and blotted to nitrocellulose for detection of the 3-B-3(−) and 7-D-4 proteoglycan populations. Two aggrecan populations are discernible. The 3-B-3 (−) CS sulphation epitope has a widespread distribution in the developing rudiment cartilage, whereas the 7-D-4 epitope has a more discrete distribution pattern in small stem cell niches in the cartilage surface. A schematic depicting a typical CS chain and digestion products generated by endoglycolytic cleavage by HYAL4 generating the 3-B-3(−) non-reducing terminal on the cleaved CS chain (**f**). Exhaustive digestion of CS by chondroitinase (Ch’ase) ABC also depolymerises the CS chain but generates a 3-B-3(+) stub epitope attached to the CS linkage attachment to aggrecan core protein. Inset image (**e**) modified from [50]. (**a**,**c**,**d**) Images supplied courtesy of Prof B. Caterson, University of Cardiff, UK. As already shown in this manuscript approximately ~1–2 in every seven non-reducing termini of CS chains in cartilage are terminated in the 3-B-3(−) epitope and these vary with age and cartilage type. The 3-B-3(−) epitope is a marker of tissue morphogenesis [36,51,52]. Stem cells are surrounded in proteoglycans decorated with this CS motif [8,9,10]. This motif is also released into synovial fluid in degenerative conditions such as OA [53,54,55,56]. Recently, Farrugia et al. [57] showed that mast cells synthesised HYAL4, a CS hydrolase that could generate the 3-B-3(−) motif in the CS chains of aggrecan and Serglycin in vitro.

**Figure 3 biomolecules-10-01244-f003:**
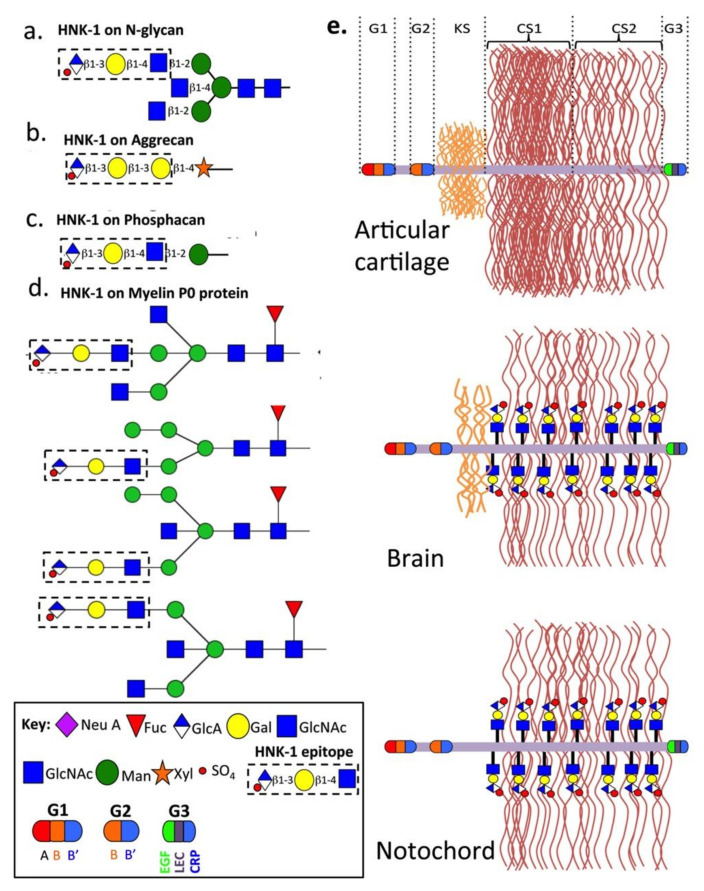
Structure of human natural killer-1 epitope (HNK-1) present on: N-glycans (**a**); notochordal aggrecan (**b**); brain phosphacan (**c**); and myelin Po glycoprotein in nervous tissues (**d**). Schematic depictions of representative aggrecan structures in articular cartilage, brain perineuronal nets and notochord in embryonic developmental tissues showing their variable relative KS contents and the presence of HNK-1 carbohydrate substitution in brain and notochordal aggrecan (**e**).

**Figure 4 biomolecules-10-01244-f004:**
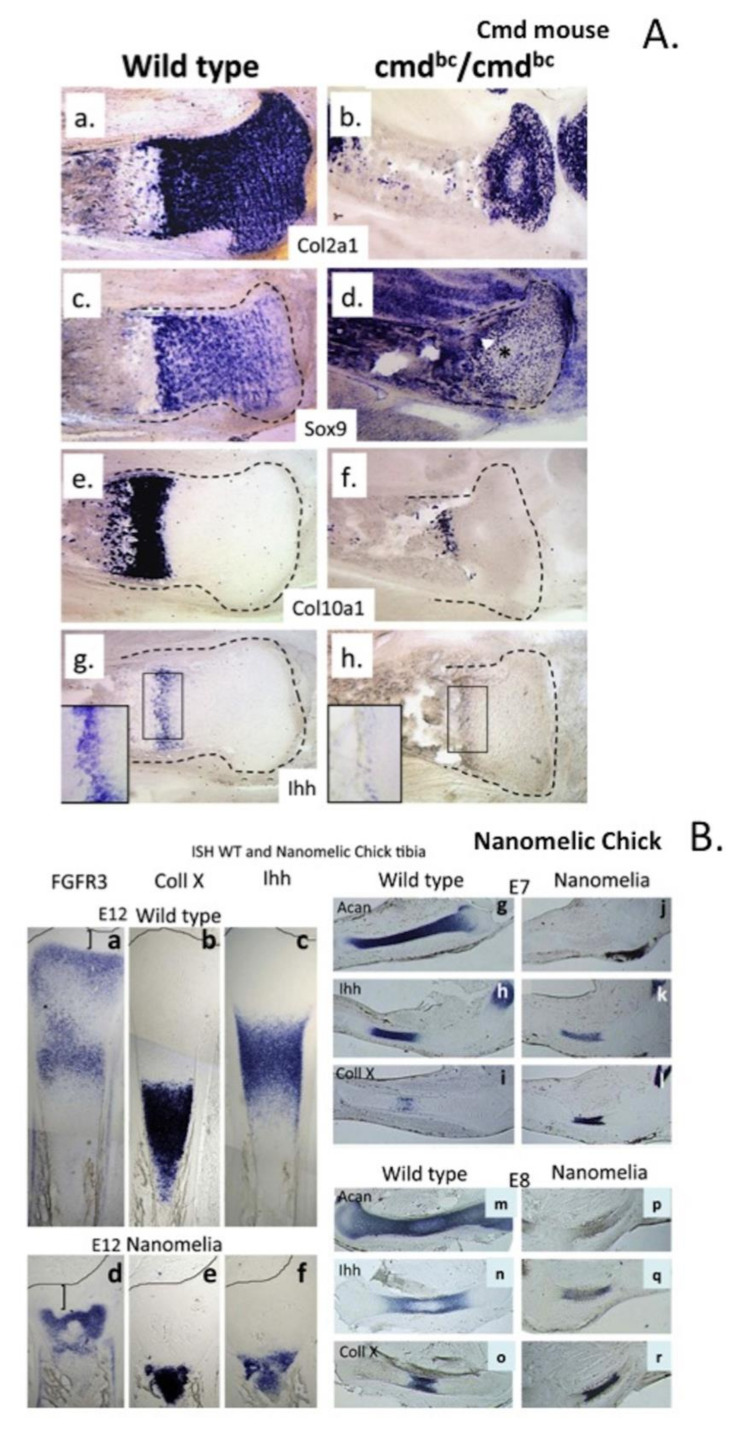
Demonstration of the modulation of growth plate cartilage morphogenesis by aggrecan in wild type (WT) (**A**) (**a**–**h**) and nanomelic E7-E12 chick tibia (**B**) (**a**–**r**). The ISH images presented demonstrate the expression of: FGFR3 (**a**); type X collagen (**b**); and Indian Hedgehog (IHH) (**c**) in WT (**a**–**c**); and nanomelic growth plate (**d**–**f**) in E12 (**a**–**f**); E7 (**g**–**l**); and E8 chick tibia (**m**–**r**). Images modified from [134] with permission using open access.

**Figure 5 biomolecules-10-01244-f005:**
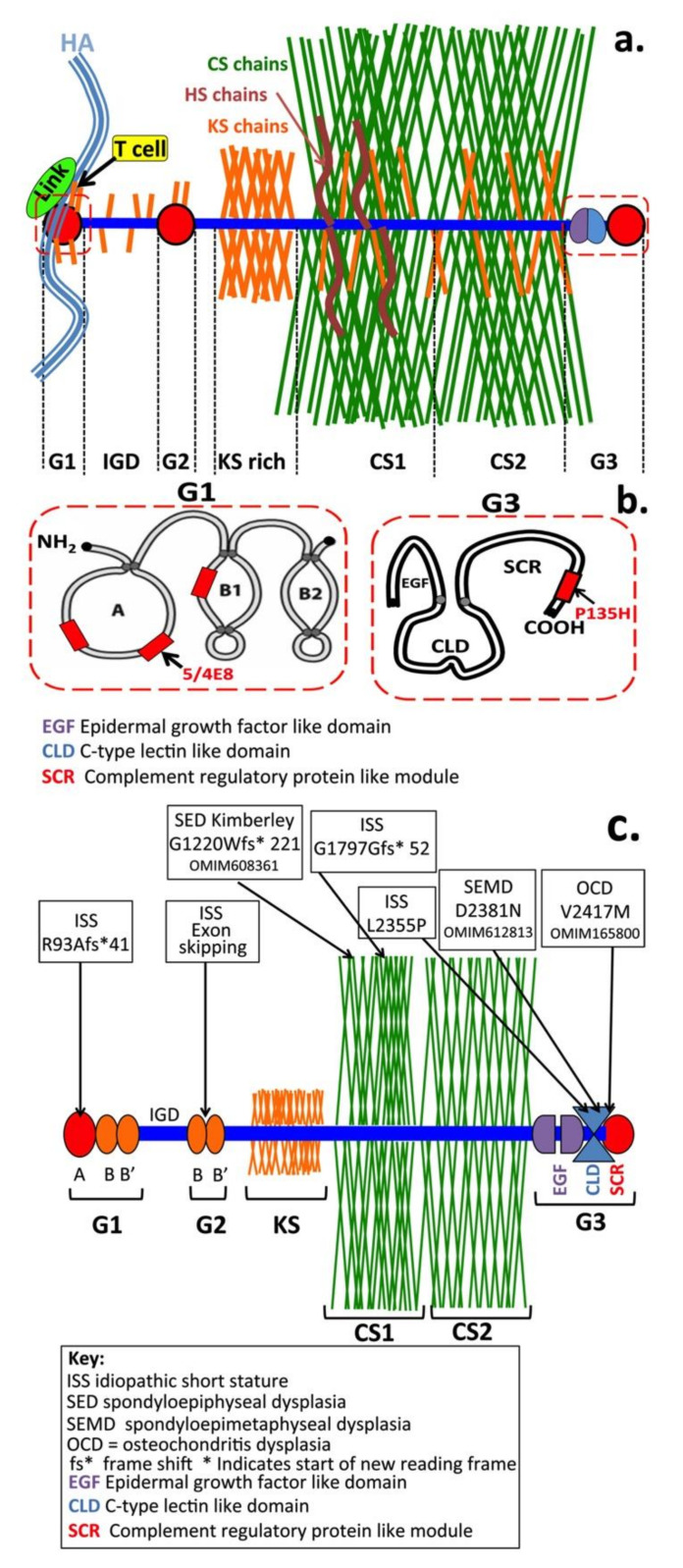
Structural organisation of aggrecan depicting the CS, KS and HS GAG chain distributions (**a**) and T cell receptor epitopes on the G1 and G3 globular domains (**b**). The aggrecanopathies showing regions of aggrecan affected by these mutations and the diseases that result (**c**).

**Figure 6 biomolecules-10-01244-f006:**
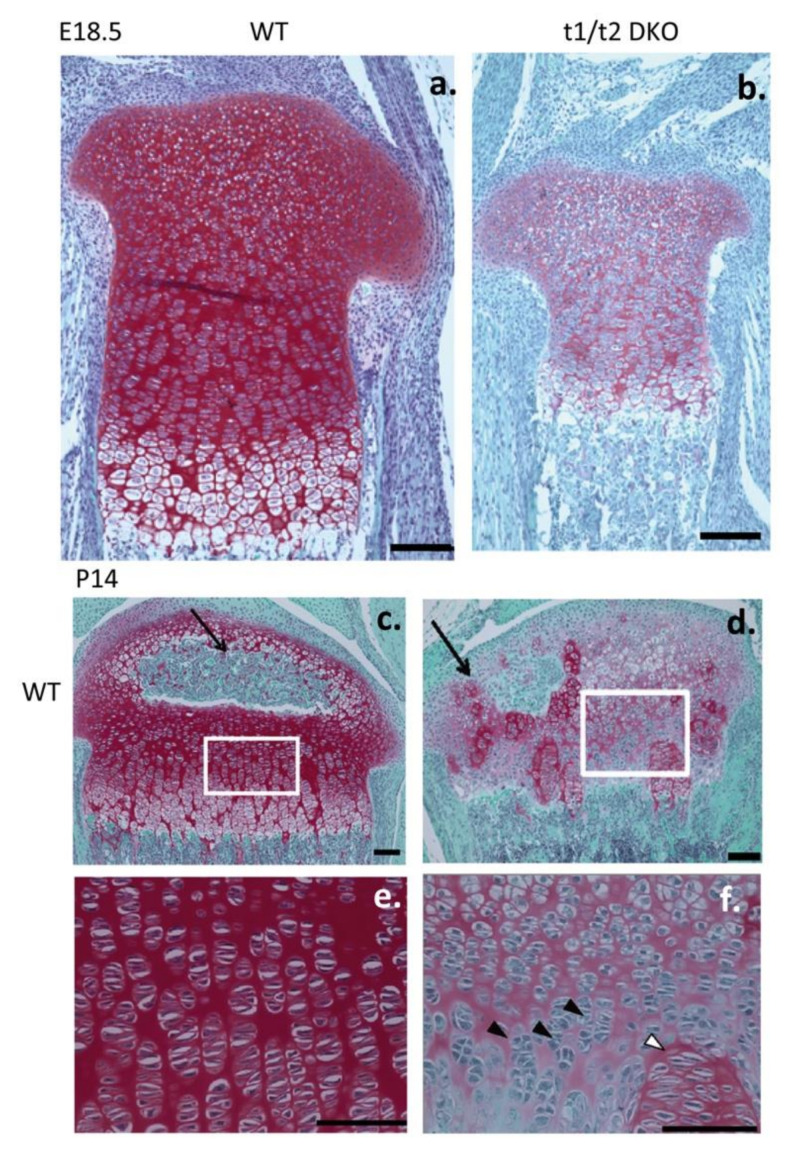
Chondroitin sulphate *N*-acetylgalactosaminyltransferase-1 and -2 (T1/2) knockout induces dwarfism in mice and altered cartilage structural organisation of the femoral condyle, its ossification centre and growth plate in wild type mice (**a**,**c**,**e**) and T1/2 knockout mice (**b**,**d**,**f**). The boxed area in (**c**,**d**) is depicted at higher magnification in (**e**,**f**). Safarin O-Fast green stain depicting aggrecan GAG distribution. Arrows depict normal ossification centre in (**c**) and abnormal structural organisation in T1/2 knockout mice in (**d**,**f**). Figure modified from [166]. Figure reproduced under the terms of the Creative Commons Attribution Licence Copyright: 2017 Shimbo et al. [166].

**Figure 7 biomolecules-10-01244-f007:**
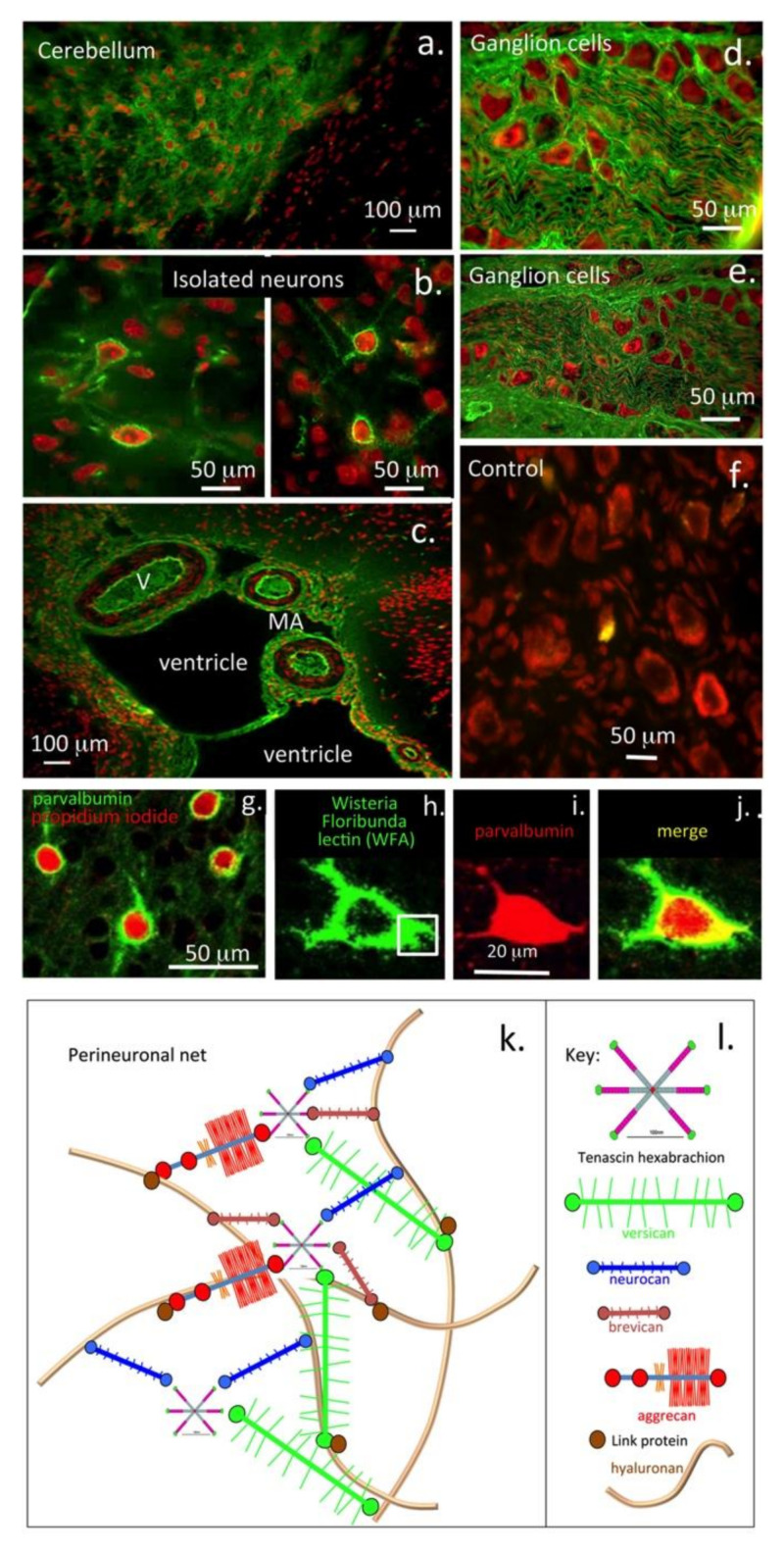
Visualisation of: perineuronal net structures (**a**,**b**); vascular features (**c**); and ganglion cells (**d**,**e**) in cerebellum and dorsal root ganglia using MAb 1B5 in confocal images. Immunolocalisation of CS Isomer 1B5 in paraformaldehyde fixed 20-μm cryo-sections of 24-month-old Wistar rat brain and lumbar dorsal root ganglia. Confocal z-stacked images of IB5 CS stub epitope generated by chondroitinase ABC digestion using Alexa 488 secondary antibody for detection and propidium iodide nuclear counterstain, mounted under coverslips using Vectasheld mountant. Images courtesy of Prof B. Caterson, University of Cardiff. Copyright Caterson, Hayes 2012 (**a**–**f**). Immunolocalisation of perineuronal nets surrounding isolated neurons in murine visual cortex using: antibody to parvalbumin (**g**,**i**); Wisteria floribunda lectin (**h**); and in a merged image (**j**). A schematic model of the perineuronal net structure in the boxed area in (**k**) showing its constituent lectican proteoglycans (aggrecan, versican, neurocan and brevican) interacting with tenascin hexabrachion and hyaluronan to form an aggregate structure stabilised by link protein. A key is provided to explain items in (**k**,**l**). Figure modified from [170] under Open Access under the auspices of a Creative Commons Attribution License.

**Figure 8 biomolecules-10-01244-f008:**
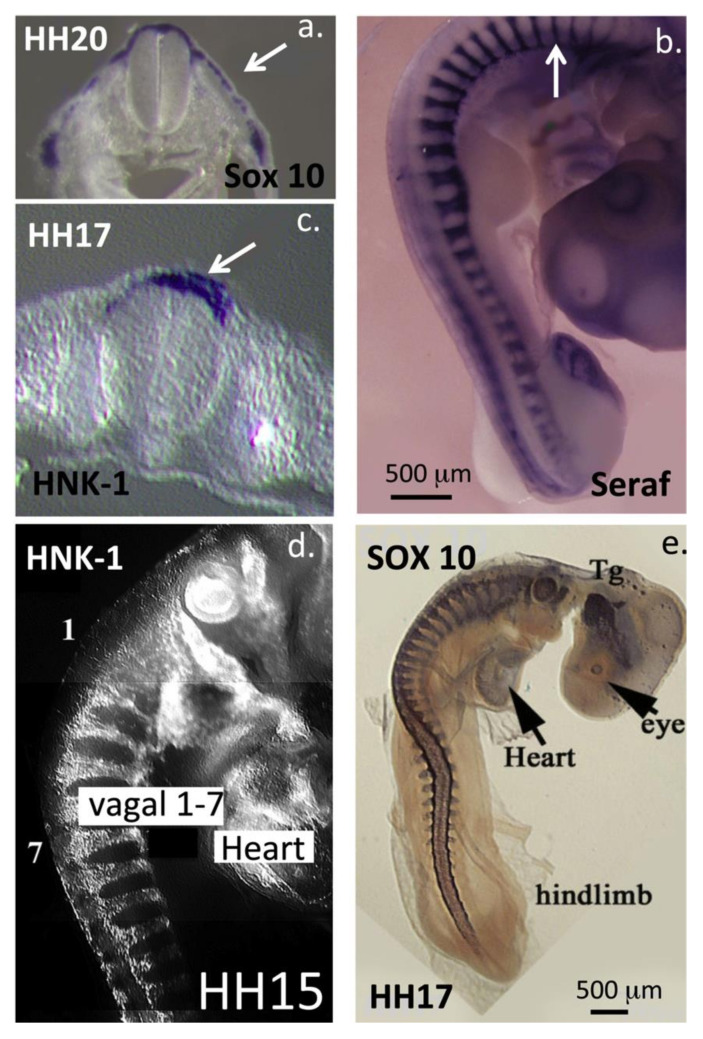
Demonstration of: Sox 10 (**a**,**e**); Seraf (**b**); and HNK-1 epitope (**c**,**d**) expression in migratory neural crest cells (**a**,**c**) and in whole mount chick embryos (**b**,**d**,**e**). (a,c,d) In-situ hybridisation images. (**b**,**e**) Immunolocalisations with specific antibodies. Seraf (Schwann cell-specific EGF-like repeat autocrine factor) is a unique protein expressed by avian embryo Schwann cell precursor cells [175]. Images reproduced from [176] under the auspices of attribution-non-commercial-no derivatives 4.0 international licence (CC BY-NC-ND 4.0).

**Figure 9 biomolecules-10-01244-f009:**
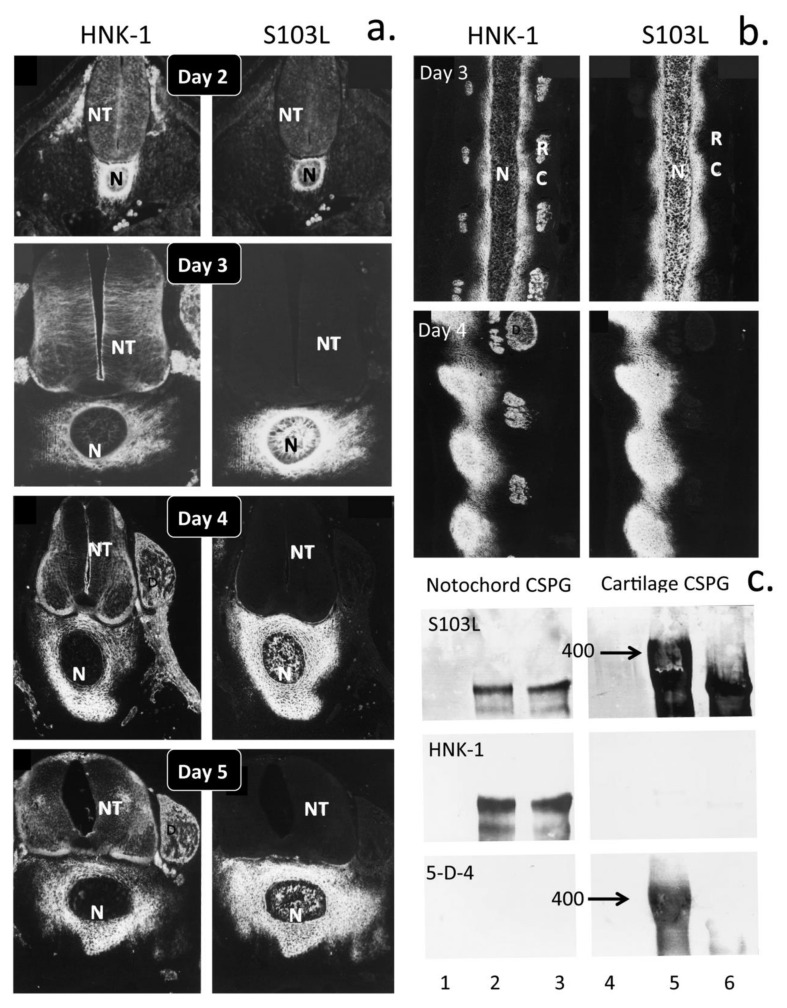
Fluorescent immunolocalisation of HNK-1 and aggrecan S103L epitope in 2–5-day-old chick trunk sections associated with the neural tube (NT) and notochord (N) development (**a**,**b**) and in Western blots (**c**) of purified chick notochordal and cartilage aggrecan. Keratan sulphate was also immunolocalised on blots using MAb 5-D-4. Notochordal aggrecan was S103L and HNK-1 positive but did not contain KS and was of a smaller molecular weight; the 400-kDa cartilage aggrecan species was not detected. Cartilage aggrecan did not stain with the HNK-1 antibody. The S103L antibody identifies the sequence ^585^XXX Glu Ileu Ser Gly Phe Leu Ser Gly Asp Arg^615^ in the CS attachment domain of aggrecan. Images reproduced from [177,178,179].

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
