# Peer review of "Aggrecan, the Primary Weight-Bearing Cartilage Proteoglycan, Has Context-Dependent, Cell-Directive Properties in Embryonic Development and Neurogenesis: Aggrecan Glycan Side Chain Modifications Convey Interactive Biodiversity"

_biomolecules, 2020, doi:10.3390/biom10091244_

Round 1
Reviewer 1 Report
Subject Appropriateness of the Manuscript:
The topic of this manuscript falls within the scope of Biomolecules
Recommendation
Accept for publication after minor revision
Comments
The paper entitled: “Aggrecan, The Primary Weight Bearing Cartilage Proteoglycan, Has Context-Dependent, Cell-Directive Properties in Embryonic Development and Neurogenesis: Aggrecan Glycan Side Chain Modifications Convey Interactive Biodiversity” (manuscript number 890040) is an interesting paper concerning the importance of aggrecan attached GAG chains modification as a functional determinant which explains diversity in aggrecans’ functional properties. A better understanding of changes in expression, disaccharide composition and sulfation patterns of GAGs during tissue repair, inflammation, and injury may be helpful in designing more effective targeted strategies in tissue repair and regenerative medicine. The manuscript shows good preparation of authors for taken theme. The great advantage of this review is the 9 detailed figures in the field of the topic, making it easier for readers who are not GAG experts. The references are chosen properly for topic of this work. Individual positions are cited correctly in the text. This review article summarizes the current state of understanding on a topic. However, some improvements should be done before eventual publication in Biomolecules. 1. The authors should attach a separate chapter describing various methods available for analysis of GAG structure. Post-translational modifications of aggrecan’s attached GAG chains can be analyzed by HPLC or capillary zone electrophoresis. However recent progress in carbohydrate chemistry has facilitated the analysis of the composition and sulfation patterns of GAGs using a FACE method. It will be interesting to describe the utility of FACE in identifying disease-related changes in GAG structure. 2. The diverse bioactivities of sulfated GAGs make these molecules an attractive class of therapeutics. More detailed information concerning the clinical utility of disaccharide composition analysis or sulfation pattern of GAGs in disease control should be provided. 3. Finally, the Authors can present the therapeutic potential of sulfated glycosaminoglycans and their synthetic mimics in the biomedical field including regenerative medicine and tissue engineering.
Author Response
Reviewer 1
I presumed you wanted a paragraph in response to each query 1-3 not a chapter since I thought the review was quite lengthy already. Added segments to the revised manuscript are highlighted.
Question 1. The authors should attach a separate chapter describing various methods available for analysis of GAG structure. Post-translational modifications of aggrecan’s attached GAG chains can be analyzed by HPLC or capillary zone electrophoresis. However recent progress in carbohydrate chemistry has facilitated the analysis of the composition and sulfation patterns of GAGs using a FACE method. It will be interesting to describe the utility of FACE in identifying disease-related changes in GAG structure.
Author response: A segment has been added covering this requested information.
Question 2. The diverse bioactivities of sulfated GAGs make these molecules an attractive class of therapeutics. More detailed information concerning the clinical utility of disaccharide composition analysis or sulfation pattern of GAGs in disease control should be provided.
Author response: A segment has been added covering this requested information.
Question 3. Finally, the Authors can present the therapeutic potential of sulfated glycosaminoglycans and their synthetic mimics in the biomedical field including regenerative medicine and tissue engineering.
Author response: A segment has been added covering this requested information.
Reviewer 2 Report
The comments are attached as a file.

Author Response
Reviewer 2.
Major points 1-8
Author response
All structures have been corrected and a revised Fig 1 and Fig 2 added to the revised manuscript.
Point 6. Some additional comments have been added to describe what is currently known about the 4C3 and 7D4 CS sulphation motifs. These comments are highlighted.
Minor points: these have all been corrected in the revised manuscript.
Round 2
Reviewer 2 Report
See the attachment.

Author Response
All requested changes have been made and new corrected Fig 1 and Fig 2 inserted into the revised manuscript.
The following segment has also been added
*As already shown in this manuscript approximately ~1-2 in every seven non-reducing termini of CS chains in cartilage are terminated in the 3-B-3(-) epitope and these vary with age and cartilage type. The 3-B-3(-) epitope is a marker of tissue morphogenesis [36, 51, 53], stem cells are surrounded in proteoglycans decorated with this CS motif [8-10], this motif is also released into synovial fluid in degenerative conditions such as OA [69-72]. Recently Farrugia et al [73] showed that mast cells synthesized HYAL4, a CS hydrolase that can generate the 3-B-3(-) motif in the CS chains of aggrecan and Serglycin in-vitro.